# Mechanisms of the Metabolic Shift during Somatic Cell Reprogramming

**DOI:** 10.3390/ijms20092254

**Published:** 2019-05-07

**Authors:** Ken Nishimura, Aya Fukuda, Koji Hisatake

**Affiliations:** Laboratory of Gene Regulation, Faculty of Medicine, University of Tsukuba, 1-1-1 Tennodai, Tsukuba 305-8577, Japan; ken-nishimura@md.tsukuba.ac.jp (K.N.); fukudaa@md.tsukuba.ac.jp (A.F.)

**Keywords:** PSC, ESC, iPSC, somatic cell reprogramming, metabolism, mitochondrion, OxPhos: glycolysis, hypoxia

## Abstract

Pluripotent stem cells (PSCs), including embryonic stem cells (ESCs) and induced pluripotent stem cells (iPSCs), hold a huge promise for regenerative medicine, drug development, and disease modeling. PSCs have unique metabolic features that are akin to those of cancer cells, in which glycolysis predominates to produce energy as well as building blocks for cellular components. Recent studies indicate that the unique metabolism in PSCs is not a mere consequence of their preference for a low oxygen environment, but is an active process for maintaining self-renewal and pluripotency, possibly in preparation for rapid response to the metabolic demands of differentiation. Understanding the regulatory mechanisms of this unique metabolism in PSCs is essential for proper derivation, generation, and maintenance of PSCs. In this review, we discuss the metabolic features of PSCs and describe the current understanding of the mechanisms of the metabolic shift during reprogramming from somatic cells to iPSCs, in which the metabolism switches from oxidative phosphorylation (OxPhos) to glycolysis.

## 1. Introduction

Pluripotent stem cells, such as embryonic stem cells (ESCs) [1,2,3] and induced pluripotent stem cells (iPSCs) [4], are characterized by their ability to proliferate indefinitely (self-renewal) and to differentiate into virtually all types of cells that comprise an organism (pluripotency). iPSCs generated from somatic cells by introduction of transcription factors greatly increased the likelihood of applying iPSCs to regenerative medicine, disease modeling, and drug development [5]. To ensure the quality and safety of iPSCs, it is critical to understand the mechanisms that underlie the reprogramming of somatic cells into iPSCs [6], which is accompanied by massive changes in gene expression and epigenetic status as well as cellular structure and functions. In addition to the epigenetic, morphological, and functional transformations, reprogramming also entails major metabolic changes as a consequence of the intrinsic differences in metabolism between iPSCs and somatic cells.

ESCs are derived originally from the cells of the inner cell mass of an embryo at the blastocyst stage. Embryos develop in a low oxygen condition before and immediately after implantation and thus rely predominantly on glycolysis for producing ATP [7]. The metabolic state of naïve and primed ESCs, which are derived from preimplantation and postimplantation blastocysts, respectively, show some distinction in accord with the drastic environmental change at the implantation [8,9]. As implantation further reduces oxygen availability, primed ESCs are even more dependent on glycolysis for ATP production than naïve ESCs. By contrast, differentiated somatic cells within the embryo reside and proliferate in a high oxygen condition, and as such, utilize oxidative phosphorylation (OxPhos) as the main source of energy production.

Metabolism supplies cells with not only metabolites for cellular functions but also energy in the form of ATP, which can be generated from glycolysis (anaerobic) and OxPhos (aerobic). Whereas OxPhos produces ATP more efficiently in terms of the number of ATP per one glucose moiety, glycolysis produces ATP more rapidly than OxPhos. Moreover, the glycolytic pathway is connected with other pathways that supply the building blocks for nucleic acids, amino acids, and lipids, which are in especially high demand in rapidly proliferating cells [10,11]. In addition to its role for efficient ATP production, the TCA cycle also supplies essential intermediate metabolites for other pathways such as lipid synthesis and epigenetic modification [9,12,13] (Figure 1). Thus, as the cells differentiate during development, they respond to the intrinsic requirements for energy and metabolites while adapting to the extrinsic environment where the cells reside. Accordingly, generation of iPSCs from somatic cells, which may be considered a reversal of differentiation, entails dramatic metabolic changes during the process [8,14,15,16,17,18,19,20].

In this review, we describe how the cells change their metabolism when they undergo reprogramming from somatic cells to iPSCs, especially focusing on recent developments that shed light on the mechanisms of metabolic changes induced by the reprogramming factors.

## 2. Metabolic Characteristics of Pluripotent Stem Cells

### 2.1. Pluripotent Stem Cells (PSCs) in a Hypoxic Environment

Mammalian eggs fertilized in the lumen of the oviduct are transported passively to the uterus, where the embryo (blastocyst) implants in the uterine wall [21,22]. The blastocyst thus develops in an environment where oxygen supply is poor [23]. Although PSCs can be cultured under high oxygen in vitro [24,25], they adapt well, or even better, to low oxygen concentration in vitro [26]. In fact, a hypoxic environment promotes expression of core pluripotency genes and enhances self-renewal and pluripotency of PSCs, partly through expression of Hypoxia-inducible factor 2-alpha (HIF2α) [7,27]. HIF2α upregulates expression of C-terminal binding proteins (CTBPs), which then promote expression of the core pluripotency factors, *Oct4*, *Sox2*, and *Nanog*, probably as a transcriptional coactivator [28]. When ESCs are adapted from normoxic to hypoxic conditions in cell culture, they switch from OxPhos to glycolysis and produce ATP anaerobically [29].

### 2.2. Glycolysis in PSCs

One common feature among ESCs and iPSCs is high metabolic flux through glycolysis [30,31,32]. Metabolite analyses demonstrate that ESCs utilize glycolysis as a main source of ATP production and produce a large amount of lactate, which is secreted into cell culture medium [16]. After the start of differentiation, ESCs diminish the high rate of glycolysis and increase OxPhos for ATP production as they differentiate into somatic cells [33,34]. The high glycolysis also shunts the metabolites through the pentose phosphate pathway in ESCs and iPSCs [35], which is important for rapid cell proliferation. Indeed, when glycolysis is inhibited by 3-bromopyruvate (3BrP), an analog of glucose that inhibits hexokinase II (HK2), ESCs undergo a metabolic switch from glycolysis to OxPhos and lose pluripotency even when cultured in the presence of leukemia inhibitory factor (LIF) [36]. Conversely, when glycolysis is maintained at a high level by overexpression of HK2 and pyruvate kinase M2 (PKM2), ESCs retain pluripotency even in the absence of LIF [37]. Thus, high metabolic flux through glycolysis is responsible for their potential for unlimited proliferation [30,32,35] and maintenance of pluripotency [36].

To maintain a high flux of metabolites in the glycolytic pathway in ESCs, the enzymes in this pathway are expressed at higher levels in ESCs than in somatic cells. Moreover, ESCs achieve the high flux of glycolysis by regulating key glycolytic enzymes including HK2, pyruvate dehydrogenase (PDH), and PKM2. High levels of HK2 and PKM2 [37,38] as well as inactive PDH [35] maintain the high glycolytic rates of ESCs (Figure 1).

Core pluripotency factors, OCT4, SOX2, and Nanog, occupy many regions of glycolytic enzyme genes and are implicated for direct transcriptional regulation of glycolysis [37]. One well-characterized core pluripotency factor is OCT4, which directly governs *Hk2* and *Pkm2* in ESCs [37]. The myc genes, *c-myc* and *N-myc*, are highly expressed in the inner cell mass of a blastocyst in vivo and also regulate self-renewal and pluripotency of ESCs in vitro [39]. Conditional knockout of both *c-myc* and *N-myc* severely compromises self-renewal and pluripotency of ESCs and results in down regulation of genes related to cellular metabolism [40]. Given the critical role for c-MYC in regulating glycolysis in cancer cells [41], ESCs also employ the *myc* genes to regulate metabolism probably by similar mechanisms to those used for maintaining rapid cell proliferation.

In addition to the core pluripotency transcription factors, a recent study showed an important role for a non-coding RNA, Lncenc1, for expression of glycolysis-associated genes [42]. Ablation of the *Lncenc1* gene significantly reduces the expression of glycolysis-associated genes and lowers glucose consumption and lactate production by over 50%, which indicates impaired glycolysis. Lncenc1 interacts with two RNA-binding proteins, polypyrimidine tract-binding protein 1 (PTBP1) and heterogeneous nuclear ribonucleoprotein K (HNRNPK), both of which regulate the expression of glycolytic genes to maintain the self-renewal ability of ESCs. Because a complex containing Lnecn1, PTBP1, and HNRNPK occupies the promoter regions of the glycolysis genes, Lncenc1, PTBP1, and HNRNPK may directly enhance transcription of these genes.

### 2.3. Structural Features of Mitochondria in PSCs

Consistent with their lesser reliance on OxPhos for ATP production, PSCs have fewer small mitochondria [31,43], as indicated by low copy numbers of mitochondrial DNA [43], and mitochondria are usually localized in the perinuclear region [30,44,45,46,47,48,49,50]. Mitochondria in PSCs also differ from those in somatic cells in their morphology and internal structure [31,35,50]. Electron microscopy shows that mitochondria in PSCs have a globular shape and their cristae are poorly developed and immature [51,52,53,54], which can be used as an indicator of high pluripotency [30,44,45,46,47,48,49,50]. Despite their lower oxidative activity, mitochondria in primed ESCs are more elongated and have more developed cristae than those in naïve ESCs [31,50,53,54,55]. When cells become terminally differentiated, mitochondria undergo further maturation to adopt more elongated and tubular morphology with numerous, highly developed cristae [35,50].

### 2.4. Functional roles for Mitochondria in PSCs

Consistent with their immature morphology, mitochondria in PSCs show lower levels of respiration and oxidative reserve capacity than those in differentiated somatic cells [30,31,32]. However, the immature and apparently underdeveloped morphology of mitochondria in PSCs does not necessarily mean that they are less functional. The importance of mitochondrial functions in PSCs [56] is corroborated by the fact that knockdown of DNA polymerase subunit γ (POLG), a subunit of mitochondrial DNA polymerase, impairs mitochondrial homeostasis and permits ESCs to lose pluripotency and differentiate [49]. In addition, ablation of growth factor erv1-like in ESCs increases expression of GTPase dynamin-related 1 (Drp1), a factor that is involved in mitochondrial fission, which then causes extreme mitochondrial fission and poor cell viability, accompanied by concomitant loss of pluripotency and impaired capacity to differentiate [57]. Thus, mitochondrial morphology reflects their essential functionality in self-renewal and pluripotency of PSCs.

Although its contribution to ATP production is low, mitochondrial electron transport chain (ETC) is fully functional in ESCs, consuming oxygen at its maximal level. Despite the maximally functioning ETC, however, mitochondrial production of ATP is kept at a suboptimal level. Uncoupling protein 2 (UCP2) in ESCs shunts pyruvate out of mitochondria, thus shifting ATP production from OxPhos to glycolysis [15]. In addition, UCP2 uncouples ETC from ATP production presumably in order to reduce generation of reactive oxygen species (ROS). OxPhos in mitochondria is known to generate ROS, which may potentially damage proteins, lipids, and nucleic acids in the cells. Because of UCP2, ESCs maintain production of ROS at a low level [48] and possess relatively low levels of oxidized proteins, lipids, and DNA [34]. The maximally active ETC in mitochondria in ESCs, although not necessarily coupled with ATP production, may be a prerequisite for rapid metabolic shift once ESCs initiate differentiation and shift to OxPhos for ATP production. In accord with this, UCP2 rapidly decreases its expression when ESCs exit from their pluripotent state [15].

Mitochondria are also important for producing metabolites that are used for purposes other than ATP production. Instead of oxidizing pyruvate completely in the TCA cycle, ESCs generate metabolic intermediates, which are then exported outside mitochondria for other purposes [58]. For example, citrate generated from mitochondrial acetyl-CoA is exported from mitochondria and then catalyzed by ATP-citrate lyase to generate cytosolic acetyl-CoA. Cytosolic acetyl-CoA is an essential substrate not only for biosynthesis of fatty acids but also for acetylation of histones [59], which is necessary for maintaining the open chromatin structure characteristic of ESCs [58] (Figure 1).

### 2.5. High Mitochondria Membrane Potential

PSCs have high mitochondrial membrane potential, which is vital for pluripotency as well as self-renewal [48,60,61,62]. ESC clones with a high mitochondrial membrane potential can differentiate into all three germ layers whereas those with a low mitochondrial membrane potential differentiate mostly into mesodermal cells [60]. Fully reprogrammed iPSCs are also observed to acquire a high mitochondrial potential [30]. Inhibition of mammalian target of rapamycin (mTOR) by rapamycin lowers the mitochondrial membrane potential, which reduces self-renewal capacity of ESCs. This high membrane potential is actively maintained by ATP synthase, which functions in reverse as ATP hydrolase using ATP derived from glycolysis. [15]. Although the role of high mitochondrial membrane potential in maintenance of pluripotency remains an enigma, it may be required for maintaining a network of fragmented mitochondria [9,63], maintaining redox potential optimal for synthesizing lipids and amino acids [64], or preparing ESCs for the energetic demands of differentiation [14]. Thus, it appears that high mitochondrial membrane potential in PSCs is not a passive consequence of low OxPhos but is actively maintained for pluripotency.

## 3. Mechanisms behind the Metabolic Shift during Reprogramming

### 3.1. Gradual Transition from OxPhos to Glycolysis

Consistent with the enhancing effect of hypoxia on reprogramming [65], PSCs rely mainly on glycolysis for ATP production, even in the presence of oxygen [53], in a manner reminiscent of the Warburg effect in cancer cells [10]. Similar to cancer cells, ESCs require high glycolysis also for shunting metabolic intermediates to various anabolic pathways [10,11,15,18] (Figure 1). Thus, the overall trend of the metabolic changes during reprogramming consists of decreasing OxPhos and increasing glycolysis [18,34], which are accompanied by alterations in the amounts of corresponding metabolites [66]. This trend of decreasing OxPhos and increasing glycolysis is amply confirmed by genome wide analyses of gene expression and protein levels as well as metabolomic profiling [30,45,66,67,68,69]. Immediately after the initiation of reprogramming, cells induce many genes relevant to metabolism and proliferation before they induce the ESC-specific genes including the core pluripotency genes [30,45,66,67,68,69]. This supports the notion that these early metabolic changes prepare cells for varying cellular demands for energy and metabolites during the subsequent process of reprogramming. The metabolic changes during reprogramming may occur gradually [30,35,70], perhaps as a response to changing demands during reprogramming. However, a series of recent findings suggest that the metabolic shift entails more complex phenomena and may even play regulatory roles during reprogramming. One such dramatic phenomena is a transient hyper-energetic metabolism, which is a hybrid of high OxPhos and high glycolysis.

### 3.2. Transient Hyper-Energetic Metabolism

Unbiased classification of the gene expression patterns during reprogramming found that metabolism-related genes show peak levels of expression at an early stage of reprogramming [68]. Consistent with this finding, a study by Prigione et al. reported that OxPhos increases as early as day 3 of reprogramming [45]. Proteomic analysis by Hanson et al. revealed a transient upregulation of mitochondrial proteins at an early stage of reprogramming [71]. Concomitant with these changes of metabolism related genes shortly after the start of reprogramming, cells undergo a transient hyper-energetic metabolism that shows characteristics of both high OxPhos and high glycolysis [72,73,74]. This hyper-energetic metabolism is somewhat reminiscent of the metabolic state observed for naïve ESCs, in which both OxPhos and glycolysis are more active than in primed ESCs [53,75]. This metabolic state of OxPhos burst generates ROS, which initiate a cascade of transcription factor induction that elicits the subsequent metabolic shift (see below). Thus, the hyper-energetic metabolism may be a regulatory cue for the overall metabolic shift during reprogramming. However, a study by Ji et al. reported that radical scavengers, N-acetyl-cysteine or Vitamin C, promote cell survival during reprogramming without altering the reprogramming efficiency and reduce DNA damage, especially de novo copy number variations [76]. This study suggests that the hyper-energetic metabolism may even be detrimental for the genomic integrity of the derived iPSCs and could be bypassed without a significant effect on the reprogramming efficiency [76]. Thus, although this hyper-energetic metabolism may accompany reprogramming, it remains to be determined if it is essential for reprogramming per se.

### 3.3. Morphological and Numerical Changes of Mitochondria

During reprogramming of somatic cells to iPSCs, mitochondria undergo a significant remodeling to adopt a rejuvenated state [30,34,35,46]. The amount of mitochondrial DNA becomes lower as reprogramming progresses toward iPSCs [77], in accord with the overall metabolic switch from mitochondrial oxidation to glycolysis during reprogramming [30,71,78]. In fully reprogrammed iPSCs, mitochondria adopt immature morphology, small in size with underdeveloped cristae, and become localized around nucleus [61].

The mechanisms behind the dramatic reorganization of mitochondria still remain elusive. However, recent reports showed that it entails mitophagy, which is selective clearance of mitochondria by autophagy (Figure 2). Autophagy of intracellular proteins or organelles is mediated by either an autophagy-related protein 5 (ATG5)-dependent or independent pathway [79]. Addition of mTOR inhibitors (i.e., rapamycin, pp242, or spermidine) to repress the mTOR signaling pathway increases the efficiency of reprogramming [80,81]. As rapamycin stimulates mitophagy via inhibition of mTOR, it implicated mitophagy as a probable mechanism for reduction in the mitochondrial content. Indeed, *Atg5*^−/−^ MEFs do not reduce the mitochondrial number upon reprogramming and fail to undergo proper reprogramming, which indicates that the ATG5-dependent mitophagy is essential for mitochondrial reduction and the progression of reprogramming [82]. Mechanistically, SOX2, one of the reprogramming factors, represses expression of the *mTOR* gene transiently at an early stage of reprogramming, and this repression of mTOR expression allows ATG5-dependent autophagy of mitochondria [82]. A more recent study, however, shows that activation of AMP-activated protein kinase (AMPK) or inhibition of mTOR may employ alternate ATG5-independent autophagy rather than canonical ATG5-dependent autophagy. This ATG5-independent autophagy appears to remove mature mitochondria as new immature mitochondria are generated during reprogramming, and this process is essential for the metabolic shift during reprogramming [77]. However, another study indicates that different reprogramming factors regulate autophagy-related genes in an opposite manner during reprogramming [83]. Thus, the mechanistic relationship between the reprogramming factors, mTOR pathway, autophagy, and mitophagy is probably more complex than currently understood and awaits further study.

In addition to the decline in the mitochondria number, reduction of the mitochondrial content during reprogramming may occur as a consequence of reduced mitochondrial size. Indeed, mitochondria appear to reduce their size by active fragmentation, which peaks 3–4 days after expression of reprogramming factors. This mitochondrial fragmentation is caused not by mitophagy but by mitochondrial fission, elicited by phosphorylation of the dynamin-related protein 1 (DRP1) GTPase, a pro-fission factor that is essential for mitochondrial fission. DRP1 is recruited to mitochondria and constricts them to elicit mitochondrial fission [84], and the activity of DRP1 is regulated positively by phosphorylation via activation of the extracellular signal-related kinase (ERK)1/2 signaling (Figure 2). During reprogramming, expression of a mitogen-activated protein kinase (MAP kinase) phosphatase, dual specificity protein phosphatase 6 (DUSP6), is reduced, which results in activation of the ERK1/2 signaling [74]. A more recent study showed that *c-myc* elicits induction of CDK1, which then phosphorylated DRP1 to promote mitochondrial fission [85] (Figure 2). This mitochondrial fission generates small mitochondria with high OxPhos ability, which likely corresponds to the hyper-energetic metabolism at an early stage of reprogramming [72,73,74].

These somewhat contradicting, but not necessarily mutually exclusive, mechanisms may relate to differences in employed reprogramming methods or multiple mechanisms that underpin the changes in mitochondrial structure and function during reprogramming. Nevertheless, these studies point to an essential, and possibly regulatory, role for mitochondrial changes during reprogramming. Indeed, the morphological differences between partially and fully reprogrammed can be quantifiable in live cells by a modified retardation modulated-differential interference contrast (RM-DIC) microscope and can be used to accurately predict the pluripotency of derived iPSCs [86]. All of these studies indicate a close relationship between remodeling of mitochondria and acquisition of pluripotency, and further study is warranted for understanding more precise mechanisms of mitochondrial remodeling and its role in reprogramming.

### 3.4. Changes in Mitochondrial Subunit Composition

Temporal proteomic profiling has revealed complex changes of mitochondrial protein levels during reprogramming. The cells at an early reprogramming stage show a complex wave of protein level changes in ETC, with decreased expression of the subunit proteins in complexes I and IV but increased expression of them in complexes II, III, and V [30]. This inevitably results in altered stoichiometry of the complexes I, II, III, IV, and V, which causes functional changes of mitochondria in addition to mere reduction of the mitochondrial content [30,71]. Given the mechanism of electron transport in ETC, low activity of complex I and high activity of complex II may favor use of flavin adenine dinucleotide (FADH_2_) over nicotinamide adenine dinucleotide (NADH) for capturing electrons, and low activity of complex IV, which converts oxygen to water, is consistent with decreased oxygen consumption [53]. High activity of complex V, namely ATP synthase, appears to promote ATP production in mitochondria, but this enzyme may also function partly as ATP hydrolase to help maintain high membrane potential [15]. Whether or not these functional changes occur simultaneously with the burst of OxPhos and the morphological changes of mitochondria is an intriguing question to be resolved, but the compositional change of mitochondrial proteins strongly suggests that cells require functional changes in mitochondria to transit through the reprogramming process.

### 3.5. Regulation of Mitochondrial Functions

The changes in the mitochondrial functions mentioned above are tightly linked to a cascade of regulation by transcription factors (Figure 2). As a part of the early changes in gene expression, the estrogen-related nuclear receptors (ERR), *Errα* and *Errγ*, as well as coactivator *Pgc-1α β* are upregulated transiently around day 3 of reprogramming at a peak of OxPhos, and ERRα/ERRγ and PGC-1α β induce the transient hyper-energetic metabolism [72]. One notable feature of this transient hyper-energetic metabolism is increased production of ROS, which was corroborated by independent studies [76,87]. The generated ROS apparently have a regulatory role and activates nuclear factor kappa B (NF-κB), activator protein 1 (AP-1), and nuclear factor (erythroid-derived 2)-like-2 (NRF2). The expressions of these transcription factors precede the peak of hypoxia-inducible factor 1-alpha (HIF1α) induction. A study by Hawkins et al. showed that, among these factors, NRF2 is required for induction of HIF1α expression because preventing NRF2 from entering nucleus by overexpression of Kelch like-ECH-associated protein 1 (KEAP1), which retains NRF2 in cytoplasm, attenuates induction of HIF1α. The induced HIF1α in turn promotes the metabolic shift to glycolysis [73]. Another study by Jang et al. showed that NRF2 is important for reprogramming because knockdown of NRF2 reduces the reprogramming efficiency. NRF2 increases proteasome activity partly via upregulation of proteasome maturation protein (POMP), which plays a key role in reprogramming [88]. It remains to be determined if a heightened proteasome activity is involved in the metabolic shift by degrading metabolism-related proteins as well.

A study by Prieto et al. showed a critical role for the endogenous *c-myc* for the metabolic shift at an early stage of reprogramming and that c-MYC plays a pivotal role for establishing a metabolic state that is high in both OxPhos and glycolysis [85], which probably corresponds to the peak of OxPhos described by another study [72] (Figure 2). The importance of c-MYC is further corroborated by the fact that c-MYC alone mimics robust OxPhos induced by OCT4, SOX2, KLF4, and c-MYC [85]. The cells with increased OxPhos and glycolysis possess high membrane potential in mitochondria, which is one notable indicator of the cells that will transit to the late stage of reprogramming and acquire the pluripotent state [85] As c-MYC and HIF1α cooperate to reprogram metabolism in cancer cells under hypoxic conditions [89,90], these transcription factors probably cooperate similarly to induce the metabolic shift during reprogramming.

In addition to c-MYC, a study by Nishimura et al. showed that KLF4 plays an important role for the metabolic shift during reprogramming (Figure 2). The effect of KLF4, however, occurs only when cells begin to acquire the fully pluripotent state because, leukemia/lymphoma 1 (*Tcl1*), one of the target genes of KLF4, is only induced at a late stage of reprogramming [78]. *Tcl1* is normally expressed only in cells of an early developmental stage including ESCs but will drive lymphomagenesis upon deregulated activation [91]. TCL1 interacts directly with mitochondrial polynucleotide phosphorylase (PnPase) and suppresses its activity [92]. PnPase is an RNAase localized in the intermembrane space of mitochondria [92] and facilitates RNA imports into mitochondria to maintain their homeostasis [93]. TCL1 induced by KLF4 or exogenously introduced TCL1 lowers the mitochondrial content and OxPhos during reprogramming [78,94]. Given its timing of induction during reprogramming, TCL1 probably lowers the mitochondrial content and OxPhos at a late stage of reprogramming [78,94].

### 3.6. Upregulation of Glycolytic Enzymes

Similar to mitochondrial genes, upregulation of glycolytic genes occurs early during reprogramming before pluripotency genes are induced [30,45,69] and continues throughout the course of reprogramming [30]. The levels of proteins involved in glycolysis also increase during reprogramming [71], and the majority of glycolytic enzymes eventually show higher expression in iPSCs than their somatic counterparts [30]. Consistent with the gradual shift from OxPhos to predominantly glycolytic metabolism, glucose usage and lactate production increase in parallel with the progression of reprogramming [30,34,35,70,78,85].

Hypoxia-related genes are upregulated during the early stage of reprogramming [45,69] (Figure 2). Concurrent with the increased reprogramming efficiency in hypoxia [69], HIF1α and HIF2α are stabilized by hypoxic conditions during reprogramming, and knockdown of HIFs in human fibroblasts prevents reprogramming [45,69], indicating the pivotal role for HIF1α in reprogramming. Just as in cancer cells [95], HIF1α promotes glycolysis during reprogramming by increasing the expression levels of the glycolysis-related genes [45,69]. The stabilization of HIFs is specific to the early phase because prolonged HIF2α stabilization into a later stage reduces the reprogramming efficiency in part via upregulation of tumor necrosis factor (TNF)-related apoptosis-inducing ligand [69].

The increase in the metabolic flow of glycolysis is controlled not just by increasing the expression levels of the enzymes but also by posttranscriptional regulation of key regulatory enzymes in the glycolytic pathway [45] (Figure 1). One crucial mechanism to regulate the relative metabolic activities between glycolysis and the TCA cycle is phosphorylation of the PDH complex, which catalyzes pyruvate into acetyl-CoA. The PDH activity is negatively regulated through phosphorylation by pyruvate dehydrogenase kinase (PDHK1) [96]. While the *Pdhk1* mRNA expression level remains unchanged, its protein level increases during reprogramming [61], probably by stabilization of PDHK1 [35]. Because the PDH complex becomes more phosphorylated and thus less active in derived iPSCs [35], conversion of pyruvate to acetyl-CoA decreases and thereby the glycolytic activity becomes higher.

Stimulation of the AKT activity increases the reprogramming efficiency [97], and consistently, modulating the activity of 3-Phosphoinositide-dependent protein kinase-1 (PDK1), which activates the AKT activity, increases the reprogramming efficiency probably through the effect on metabolism [98] (Figure 3). For example, PDK1 activator, PS48 [99] promotes reprogramming whereas its inhibitor, UCN-01 [100], decreases the reprogramming efficiency through their effect on the AKT activity. In these studies, the AKT activity was shown to correlate with increased expression of the glycolysis-related genes and higher lactate production [98].

Pyruvate kinase catalyzes conversion of phosphoenolpyruvate to pyruvate, which is the final step of glycolysis (Figure 1). One of the four pyruvate kinase (PK) genes, the PKM gene, produces two isoforms PKM1 and PKM2 via alternative splicing [101]. The shift from PKM1 to PKM2 promotes glycolysis because PKM2 has a lower catalytic activity than PKM1. This switch from PKM1 to PKM2 occurs during reprogramming and contributes to the enhanced glycolytic activity [61].

At a late stage of reprogramming, induction of *Tcl1* by KLF4 plays an additional role for promoting the metabolic shift [78]. As a coactivator of AKT, TCL1 enhances the AKT activity, which further elevates some key glycolytic enzymes. Exogenous expression of the oocyte factors TCL1 or TCL1b1, a protein related to TCL1, also increases the reprogramming efficiency by promoting the metabolic shift at this stage [78,94], which occurs independent of changes in cell proliferation [94]. TCL1 is shown to promote phosphorylation of AKT by direct interaction and its polymerization [102] (Figure 3). This mechanism may function cooperatively with the PDK1-mediated phosphorylation of AKT to further enhance the AKT activity that has already become high during the preceding stages of reprogramming.

## 4. Perspectives

Although the metabolic shift has been documented in many reprogramming systems and occurs throughout the reprogramming process, its molecular mechanism is only beginning to be deciphered. As outlined in this review, modulating the metabolism in general impacts the efficiency of reprogramming, and conversely, when the reprogramming efficiency is altered, it is usually accompanied by an altered metabolism. This suggests that metabolic shift may not be a mere byproduct of the reprogramming process, but rather plays a more active or regulatory role in this process.

Although recent studies clarified the general trend of the metabolic shift during reprogramming, analysis of metabolic regulation is still faced with challenges because the metabolic system is tightly linked to complex networks of feedback and feedforward loops, both at the transcriptional and posttranscriptional levels, to maintain homeostasis. These regulatory networks render interpretation of the results difficult, especially with regards to causality. In addition, reprogramming progresses asynchronously in a heterogeneous population of cells, and individual cells may take different paths when they transit from one metabolic state to another. Thus, analyses using a population of cells may only provide an average picture of events, in which details may be blurred or escape detection. Furthermore, the sequence of events during the metabolic shift may depend on, and therefore differs among, the employed reprogramming system.

One important difference in the reprogramming system is the use of human or mouse cells. In general, human ESCs are more similar to primed mouse ESCs rather than to naïve mouse ESCs. Because it is often difficult to determine if derived human or mouse iPSCs in the literature correspond to the naïve or primed state, this review did not mention the differences in the metabolic states between naïve and primed ESCs. For the detailed metabolic differences between naïve and primed ESCs, which are also relevant to the differences between human and mouse iPSCs/ESCs, we refer readers to excellent reviews on this topic [8,103].

Despite these difficulties of investigating the metabolism during reprogramming, recent developments of analyzing gene expression [104] and metabolomics at a single-cell resolution [105] provide an exciting possibility of understanding the metabolic shift and its complex regulations in unprecedented clarity and detail. Understanding the metabolic basis of stemness, both in embryonic and adult stem cells, should provide crucial information for applications of stem cell therapies in the future.

## Figures and Tables

**Figure 1 ijms-20-02254-f001:**
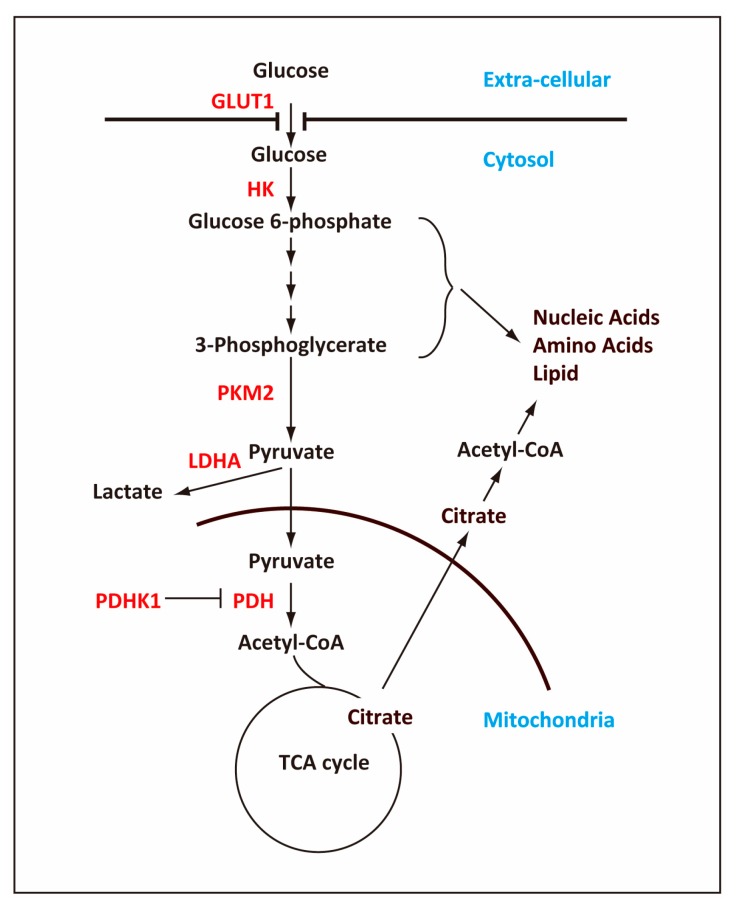
Outline of the glycolytic pathway and TCA cycle. The cell uses glucose transporter, GLUT1, to take up glucose, which is metabolized ultimately to pyruvate in the glycolytic pathway, producing 2 ATP molecules per one glucose molecule. In pluripotent stem cells (PSCs), a majority of glucose-derived pyruvate is converted to lactate and secreted out of the cell while some of pyruvate is transported into mitochondria and converted into acetyl-coenzyme A (acetyl-CoA). The pathways that are connected with the glycolytic pathway produce ribose, nicotinamide adenine dinucleotide (NADPH), and amino acids, which are required for rapidly proliferating cells such as PSCs and cancer cells. In mitochondria, acetyl-CoA is converted into citrate, which, in addition to oxidization in the TCA cycle, will be transported out of the mitochondria into the cytosol and converted back into acetyl-CoA. The cytosolic acetyl-CoA is important for reactions such as lipid synthesis and histone acetylation. Key enzymes that are regulated in the glycolytic pathway are indicated in red. Arrows indicate the flow of metabolites, and the T bar indicated negative regulation of PDH by PDHK1.

**Figure 2 ijms-20-02254-f002:**
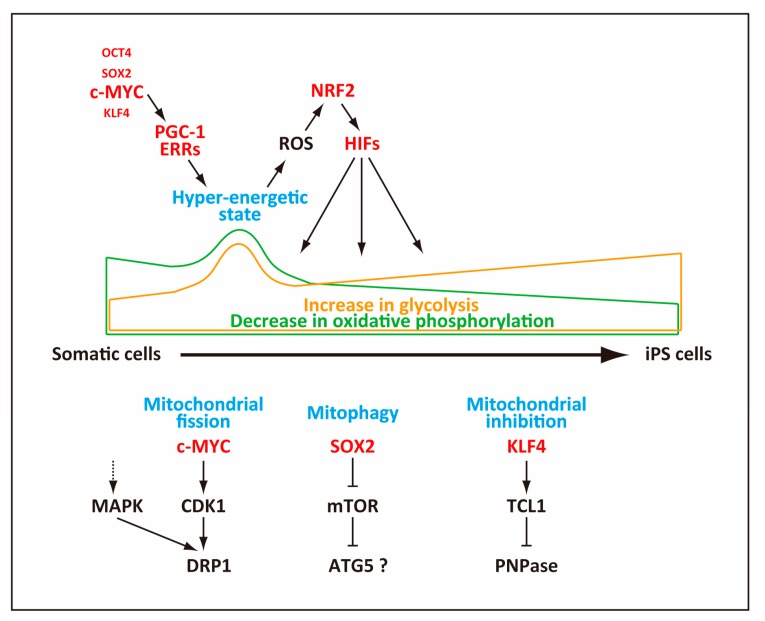
Mechanisms of mitochondrial changes during reprogramming. Mitochondria contribute to the hyper-energetic metabolism that generates reactive oxygen species (ROS) at an early stage or reprogramming. The generated ROS serve as a signal to activate nuclear factor (erythroid-derived 2)-like-2 (NRF2), which then induces HIFs. Mitochondria also undergo fission, autophagy (mitophagy) as well as functional inhibition, ultimately becoming less active in producing ATP.

**Figure 3 ijms-20-02254-f003:**
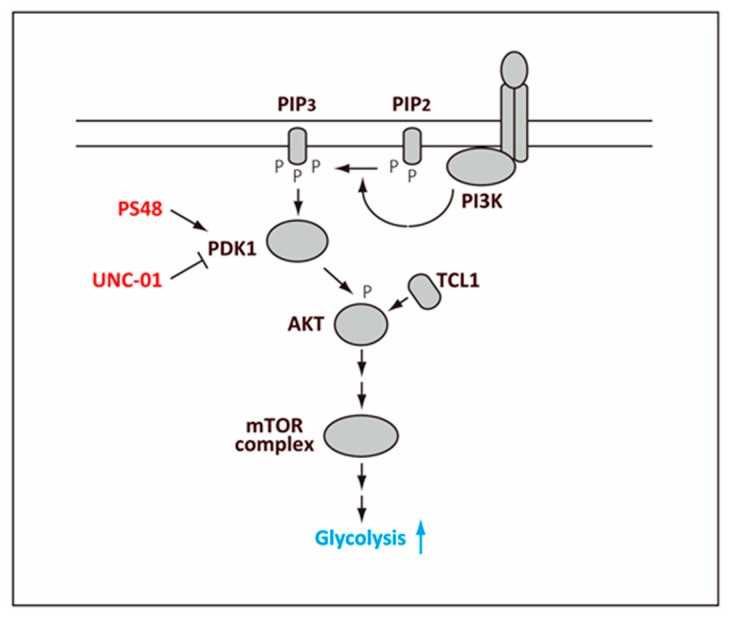
Regulation of glycolysis by AKT. Phosphoinsitide 3-kinase (PI3K) associated with a receptor, such as a receptor tyrosine kinase, phosphorylates phosphatidylinositol (4,5)-bisphosphate (PIP2) to generate phosphatidylinositol (3,4,5)-triphosphate (PIP3). PIP3 then binds and activates 3-Phosphoinositide-dependent protein kinase-1 (PDK1), which then activates AKT by phosphorylation. The phosphorylated AKT elicits activation of the mTOR complex, which leads to higher glycolysis. Leukemia/lymphoma 1 (TCL1) binds to AKT directly and promotes phosphorylation of AKT.

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
