# Peer review of "Mechanisms of the Metabolic Shift during Somatic Cell Reprogramming"

_ijms, 2019, doi:10.3390/ijms20092254_

Round 1
Reviewer 1 Report
Mechanism of the Metabolic Shift during Somatic Cell Reprogramming
This review discusses the current knowledge on metabolic changes during conversion of somatic cells to pluripotency. Somatic cell reprogramming is a complex process that yields pluripotent stem cell-like cells that differ in their developmental potential. To improve the quality of IPSCs , and to address fundamental questions about control of cell identity, molecular mechanisms of the reprogramming process must be understood. Here the authors discuss recent discoveries regarding the mechanisms of the metabolic shift during reprogramming from somatic cells to iPSCs, specially the metabolism switches from OxPhos to glycolysis, and specially the role of mitochondria.
Comment:
The review is well written, comprehensive and up to date.
Suggestion: If there are any differences in metabolic changes between mouse and human ipSC or ESC reprogramming, that should be pointed out in a small paragraph.
Author Response
As reviewer suggested, we have added a paragraph in the Perspectives, and included a reference to a review article that summarizes these differences.
We have omitted the description human and mouse cells, as it is difficult to see from the published articles if there is any species-specific differences in metabolism. However, most of human iPSCs/ESCs are equivalent to mouse iPSCs/ESCs in the primed state, and in this sense, there are metabolic differences between human (mostly primed, epiSCs) iPSCs/ESCs and mouse (primed and naïve) iPSCs/ESCs.
In fact, at the time of writing/revising first drafts of this review, we decided to remove the description of the metabolic difference between the naïve and primed states, which was initially included in this review. This is because the state of iPSCs (in the naïve or primed state) was unclear in most of the published articles on metabolic changes during reprogramming. This omission, however, made the review far more easy to follow and more focused on the mechanistic aspect of the metabolic changes that are common to both types of iPSCs/ESCs.
Reviewer 2 Report
This review article nicely summarized recent knowledges concerning metabolic features of pluripotent stem cells and metabolic changes during somatic cell reprogramming, as well as their roles and molecular mechanisms of their regulation. The manuscript is written well and is clearly presented.
Specific comments;
1. Please cite figures at appropriate positions in the text.
2. Additional schematic figures showing the AKT related pathways (lines 350-370) should be helpful to understand the described molecular linkages.
Author Response
1. Please cite figures at appropriate positions in the text
We have added to citations to Figure 1 and Figure 2, as well as to the newly added Figure 3 at the end of corresponding sentences.
2. Additional schematic figures showing the AKT related pathways (lines 350-370) should be helpful to understand the described molecular linkages.
We have added a new figure as Figure 3, which depicts the outline of the AKT signaling pathway. The Figure is also referred to in the appropriate positions in the text.